# Efficient training approaches for optimizing behavioral performance and reducing head fixation time

Anna Nasr[1]*, Sina E. Dominiak[2], Keisuke Sehara[3], Mostafa A. Nashaat[1], Robert N. S. Sachdev[1], Matthew E. Larkum[1]*

1 Institut für Biologie, Neurocure Center for Excellence, Chariteplatz 1 / Virchowweg 6, Charité Universitätsmedizin Berlin Humboldt Universität, Berlin, Germany, 2 University of Sussex, School of Life Sciences, Brighton, United kingdom, 3 Department of Physiology, Graduate School of Medicine, The University of Tokyo, Tokyo, Japan

* anna.nasr87@gmail.com (AN); matthew.larkum@gmail.com (MEL)

**Data Availability Statement:** All written Software is now available for download from: https://gin.g-node.org/nasra/Prior-experience-accelerates-

## Abstract

The use of head fixation has become routine in systems neuroscience. However, whether the behavior changes with head fixation, whether animals can learn aspects of a task while freely moving and transfer this knowledge to the head fixed condition, has not been examined in much detail. Here, we used a novel floating platform, the "Air-Track", which simulates free movement in a real-world environment to address the effect of head fixation and developed methods to accelerate training of behavioral tasks for head fixed mice. We trained mice in a Y maze two choice discrimination task. One group was trained while head fixed and compared to a separate group that was pre-trained while freely moving and then trained on the same task while head fixed. Pre-training significantly reduced the time needed to relearn the discrimination task while head fixed. Freely moving and head fixed mice displayed similar behavioral patterns, however, head fixation significantly slowed movement speed. The speed of movement in the head fixed mice depended on the weight of the platform. We conclude that home-cage pre-training improves learning performance of head fixed mice and that while head fixation obviously limits some aspects of movement, the patterns of behavior observed in head fixed and freely moving mice are similar.

## Introduction

Head fixation has been used in neuroscience for almost 60 years, beginning with the work of Jasper, Evarts, Lemon and colleagues [1–3]. These seminal experiments initiated a decades-long investigation into the relationship between activity in sensory-motor cortices and behavior, and established head fixation as a standard technique for combining awake behavior and sophisticated recording and imaging approaches [4–8] Head fixation has some advantages: it prevents head movement, reduces movement artifacts, reduces available responses of the animal, and makes behavioral tracking offline and in real-time easier [9–13]. It also makes possible to acutely position recording electrodes [1–3]. However, head fixation typically comes with a great price. Many behavioral paradigms, including navigation in an environment [14],

training-of-head-fixed-mice-on-a-floating-platform.
git" the data behind the figures is also uploaded to
gin.

**Funding:** The following funding sources have
supported this project: (1) Deutsche
Forschungsgemeinschaft (DFG), Grant Nos.
246731133, 250048060 and 267823436 to ML; (2)
DFG Project number 327654276 – SFB 1315 to
ML; (3) European Commission Horizon 2020
Research And Innovation Program and Euratom
Research and Training Program 2014–2018 (under
grant agreement No. 670118 to ML); (4) Human
Brain Project, EU Commission Grant 720270
(SGA1), 785907 (SGA2) and 945539 (SGA3) to
ML; (5) Einstein Foundation Berlin EVF-2017-363
to ML. The funders had no role in study design,
data collection and analysis, decision to publish, or
preparation of the manuscript.

**Competing interests:** The authors have declared
that no competing interests exist.

are not possible or are severely compromised by the inability of the animal to move freely, especially by the limitations imposed on head movement. Additionally, head fixation limits the field of view and reach of the animal. This restriction can be used in the behavioral monitoring to narrow down the modalities that animals are trained to use: limbs, eyes or whiskers. The downside is that it can also increase the time for animals to learn a new behavior, in part because they have to adjust to head fixation, and they have to learn to move and behave while head fixed. Nevertheless, head fixation is still the method of choice for a wide variety of experimental approaches with awake behavior in rodents, including behavioral tracking, multiphoton imaging, widefield imaging and intracellular recording techniques. The limits placed on natural behavior by head fixation has motivated the development of alternative methods, including miniaturized, head mounted approaches [15–17]. For many years, head mounted approaches were restricted to extracellular recordings with tungsten electrodes [18], however more recent studies include one- and two-photon imaging techniques ([19–22]) and even whole-cell recordings [8]. Head mounted methods frequently involve "tethering" the animal via wires or optical cables, which can be alleviated using wireless approaches. Despite these advances, head mounted devices typically come with one or more compromises in terms of the quality and flexibility of the recordings and therefore have been mostly employed with electrophysiological, and not with modern imaging methods. Another approach has been to increase the dimensions of behavior available under head fixation, for example virtual reality approaches have been established [23, 24]. Virtual reality allows a head fixed animal to explore an almost natural environment that is controlled in a closed-loop situation where the movement of the animal determines the scenes displayed on monitors in the visual field. This powerful new development of whole environments around head fixed mice has focused almost exclusively on visual information. A second related development has been to use "real world" floating-platform environments that mice navigate while head fixed ([9, 25, 26]). Floating platform approaches are well-suited to tactile and multi-modal behaviors. Animals are able to "walk" through a physical, real-world environment while head fixed. The frictionless platform moves under the animal as it walks. This enables the most advanced recording methods to be used in combination with movement in a real-world environment. But in both virtual and real world environments training mice can be challenging. The fundamental issue which we tried to address here was to improve the learning rate of mice and in the course of doing so we also examined how and whether head fixation changed behavior. Our work suggests that pre-training animals under freely-moving conditions reduced the total time the animals spent learning while head fixed and that head fixation does not trigger aberrant behavior.

## Materials and methods

All experiments were conducted in accordance with the guidelines of animal welfare of the Charité Universitätsmedizin Berlin and the local authorities (Landesamt für Gesundheit und Soziales, LAGeSo). To minimize the time between home-cage training and Air-Track training as well as outside factors, all mice underwent surgery at the beginning of the experiment.

### Surgery

Adult C57BL / 6 mice (n = 12) were anesthetized with Ketamine/Xylazine (95 mg / kg and 5 mg / kg), positioned in a stereotactic apparatus (David Kopf Instruments, California, US) and placed on a heating pad (FHC Inc. Maine US). During surgery, the eyes were covered with ointment (Bepanthen, Bayer, Leverkusen, Germany) to prevent them from drying out. Before making an incision, Lidocaine was injected under the skin and the scalp was disinfected with ethanol. The skull was exposed, the fascia on the bone scraped off with a dental scraper and the

skull was air dried. A light-weight aluminum head post was laid on the skull and RelyX (3M, Minnesota, US) cement was used to affix the head post to the skull ([9, 25, 27]). Black jet acrylic (Ortho Jet, Lang Dental) was used as a second layer to cover the exposed bone and to enhance the cementing of the head post. For analgesic purposes, Buprenorphin (0,05–0,1 mg/kg) was injected IP on the day of surgery and Carprofen (5 mg/kg) was injected IP on the day of surgery and for two days post surgery.

## Air-Track

The Air-Track platform utilizes a simple design with wide experimental potential ([25]). In this report, we used a clear plexiglass air table mounted on aluminum legs (Fig 1A). Pressurized air (300 kPa, 45 psi) was pushed out from 1 mm holes that were spaced 8 mm apart. This provided jets of pressurized air under a 3D printed platform. The platform that rested on the air table was round, 15 cm in diameter and weighed 75g. A set of walls attached atop the platform were shaped into a Y. Each lane of the maze was 7 cm long, 4 cm wide and 3 cm high (Fig 1A). The Pixy camera—Arduino interface tracked the position of the maze at 35 fps temporal resolution ([25]). This interface was used to determine the location of the maze with respect to the mouse, and was used to trigger an actuator that moved the reward port toward the animal when it entered the correct lane.

## Daily training routine: Common to both home-cage and head fixed training

Mice were housed in a reverse day night cycle regime and handled twice a day in their active period, in the dark cycle. For home-cage training, mice were left in the specially designed

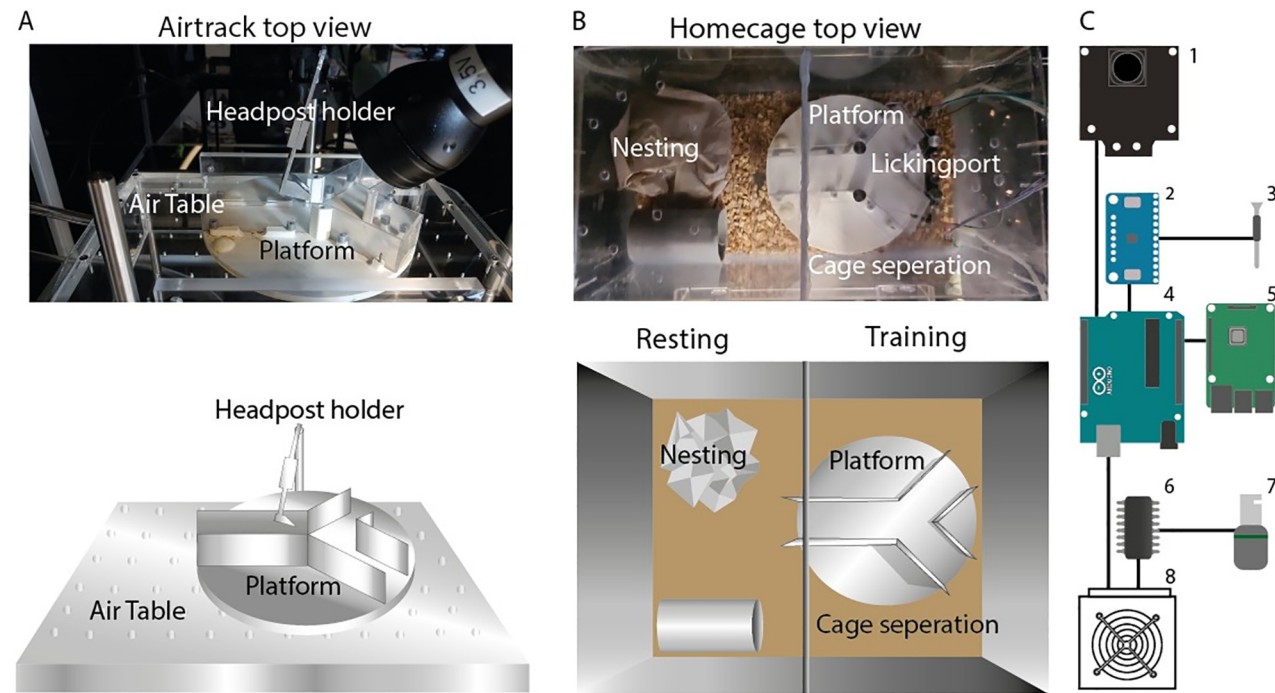

**Fig 1. Air-Track with automated home-cage training system.** A) The Air-Track with the Y maze seen from a top view. The maze is 15 cm in diameter and 3 mm thick. The platform can be easily moved from the Air-Track to the home-cage. B) Top view of the standard home-cage with the Y maze and the separation of resting and training area. C) Simple schematic of the electrical components and their connections: 1) Pixy cam, 2) Touch capacitor, 3) Licking port, 4) Arduino uno, 5) Raspberry Pi, 6) H bridge, 7) Solenoid, and 8) Power supply.

home-cage to explore the apparatus in their active time (Fig 1B). All the electrical components related to behavioral control were housed in a box outside the home-cage, with only the licking ports entering the y maze (Fig 1C). When training in the home-cage began, mice were put on a water restriction regime. Their weights were monitored daily and water intake and training schedules were adjusted to maintain each animal's weight at 85% of the starting weight ([28]). In the first few days of training, mice were guided to the lick ports with drops of water, until they learned that the lick ports delivered reward. One practical issue in the home-cage, from a measurement and monitoring perspective, was that if mice were trained with ad-lib reward, they tended to spread out successful trials over 24 hours. Mice attempted more trials in this condition, but also made many more errors ([29–32]). To reduce variability in timing and con- text of trials, mice were put on a twice daily training schedule. In the home-cage, water was only available during these two training sessions. One session began at the onset of the night (active) cycle and one at the end of the night cycle. The beginning of a training session was indicated by both light and sound cues.

**Home-cage training.** Mouse training was separated into two parts. One part was the float- ing platform, the "Air-Track" (Fig 1A) and the second part was the home-cage training system (Fig 1B). Training of animals in home-cage began after a head post had been surgically implanted. They were first habituated to being handled, then they were habituated to the appa- ratus, and when they were ready for the Air-Track, they were habituated to having the head post held. The home-cage was a standard mouse cage separated into two areas, the resting area in which we placed nesting and food, and a training area which had a Y-maze identical to the one that was used in the head fixed condition on the Air-Track. In the home-cage, there were two Arduino (Fig 1C4) gated licking ports (Fig 1C3) fixed in place at the end of each lane of the Y maze. In the Air-Track, there was one lick port which descended into a lane. Lick ports were activated when the tongue contacted the steel nozzle of the licking port. Contact triggered the touch capacitor (Fig 1C2) the solenoid (Fig 1C6/7) to open for 0.2 s, dropping 8μl of reward. In the home-cage, a Pixy camera (Fig 1C1) tracked each animal as it moved into the center of the maze. When mice were housed socially, color tracking of the head posts was used to distinguish animals from each other ([25, 33]). The same Pixy camera / Arduino interface was used in both the Air-Track and Home-cage, while in the Home-cage the Pixy camera was positioned above the animal- In the Air-Track, the Pixy camera was placed under the platform. When mice reached a stable 80% correct level in the Home-cage (1 week), they were transferred to training on the Air-Track, where they were first habituated to head fixation.

**Habituation to head fixation.** The mice were acclimated to being handled, during which the animals obtained a reward while the head post was manually held in place by the experi- menter. This process was repeated multiple times in the first 3 days of habituation. On the fourth day, mice were head fixed on the platform for the first time. The duration of head fixa- tion was increased daily by 10 minutes each day, till mice were head fixed for 30 minutes. Dur- ing this habituation phase, mice were periodically rewarded with a condensed milk reward, and were monitored for signs of stress i.e. vocalization or excessive movement. Mice that showed no evidence of stress were then trained in the task. Their weights were monitored daily. Weights were not allowed to fall below 85% of their initial weight. Training for the task started 3 days after the water restriction paradigm was initiated. Water restriction was neces- sary to motivate the mice to perform in the experiment. When mice were being trained, they acquired their entire complement of water in the training time. If mice lost more than 15% of their initial weight and failed to obtain enough water during training, they were given addi- tional water, 2–3 hours after the experiment.

### Tracking the mice in the home-cage

We use the Pixy camera (CMUcam5 Image Sensor) (Fig 1C1) to track the location of mice. The Pixy camera can use both a frame by frame mode, and a coordinate system which presents the position with an X and a Y value. By laying this coordination system over the home-cage we can monitor the real time position of each mouse in the Home-cage. The output of the Pixy camera was sent to an arduino (Fig 1C4) and in addition the information from the touch capacitor was sent to a computer or a raspberry-pi (Fig 1C5). The output was used offline for estimating the position and performance of each mouse, at each time point. While other options are available for tracking mice, like RFID chips, we chose Pixy cam for its low cost, and ease of use. In addition, the Pixy cam works for tracking freely moving mice in the home-cage, and for tracking the position of mouse in the Air-Track. In our hands, with the Pixy cam it was possible to use the same program with minor tweaks in both setups.

### A trial in home-cage

When a mouse entered the maze, the identity of the mouse was checked via the color tracking algorithm and Pixy camera (see below) deployed to track colors (under visible light) painted on the head post of each animal. Once the ID had been established, one of the two lanes of the Y maze were activated by the Arduino. This was indicated by activating both a LED and buzzer associated with the lane. The trial could end in one of three ways: 1) a successful trial where the mouse licked the correct licking port; 2) a failed trial, where mice entered incorrect lanes or licked the wrong lick port; 3) an aborted trial, which occurred if the mouse exited the maze without entering a lane or licking a lick port (Fig 2A).

### A trial in the Air-Track

When mice were head fixed, they were trained in the same two choice paradigm that was used in the home-cage (Fig 2B). Mice were trained to recognize the direction of the auditory / visual stimulus and were expected to turn the maze in the correct direction and enter the correct lane. One difference between the freely moving and head fixed condition was that In the head fixed condition, tracking involved monitoring the maze with the Pixy camera, not monitoring the mouse or the mouse ID. In the Air-Track, an active lane was designated by our in house custom software that tracked and monitored the trial by trial performance of each mouse. The output of the Pixy camera was a set of coordinates sent to an Arduino. These coordinates were converted into the real time position of the mouse in the maze. At the outset of training, for the first 3–4 days, each trial was guided manually by the experimenter. For each trial, the platform was nudged in order to evoke forward and backward movement from the mouse.

### Monitoring behavior and correcting for bias

A custom in house code was designed for training and data acquisition. The code consisted of two parts: the first was the basic paradigm (Fig 2) and the second was an adjustment for biases (Fig 3A). With only 2 lanes in the discrimination task, it was not possible to completely randomize the task. To prevent too many similar trials in a row, the distribution of active lanes selected for each mouse, on each day was monitored. If in a single session, a lane was activated more than 60% of the time, or a mouse in the next trials and sessions, the adjacent lane was activated more often. This procedure helped maintain an average of 50% activation for each lane. Bias correction: Previous work has shown that mice develop a bias in a 2-choice paradigm. They learn to obtain all their reward by simply performing correctly on half the trials. To counter this, many paradigms use bias correction ([28]). In our case, we monitor the

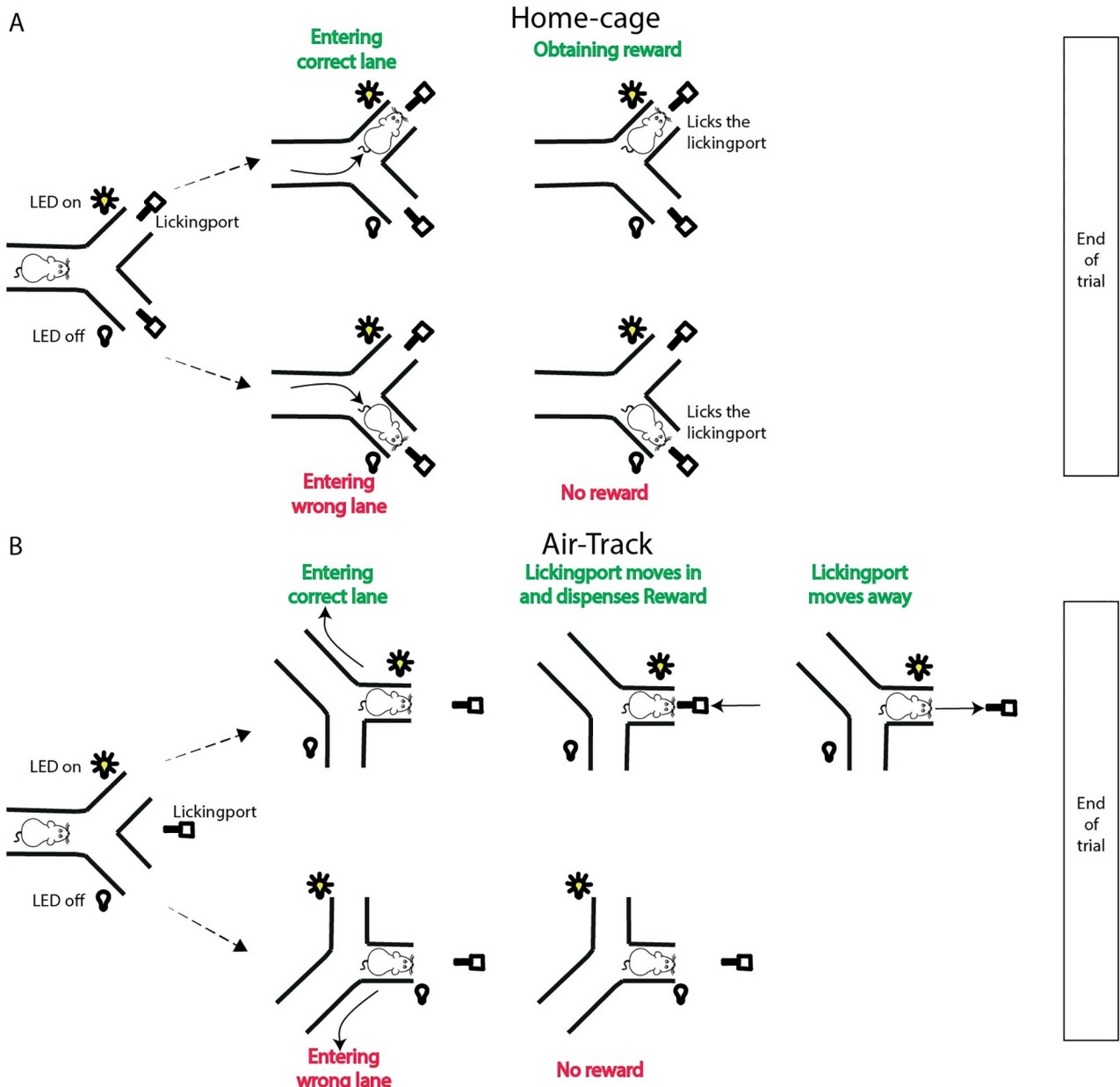

**Fig 2. Schematic representation of the outcomes from each trial.** A) Mice in the home-cage entered the choice-point in the middle of the maze, and waited for a lane to be activated (indicated by a LED). If the mouse entered the active lane and licked at the licking port, a reward was given. If the mouse licked at the incorrect licking port in the wrong lane, no reward was given. In both cases, the lights turn off, and a new trial begins when the mouse has moved back to the choice point in the maze.B) In the Air-Track, the licking port only approaches when the mouse is in the correct lane. The choice of lane, not the licking at the licking port, determines whether the mouse gets a reward. A Pixy camera was used to track the angular and XY position of the mouse in the maze.

distribution of lanes used by each mouse, in each session. When bias was detected, the lane that the mouse prefered was deselected, and the non-preferred lane was activated more often (Fig 3B). If this mild adjustment was ineffective, a stricter bias countering method was initiated. The non-preferred lane was activated 5 times in a row and a new bias counter was initiated, until mice obtained a reward 3 times in a row in the non-preferred lane (Fig 3C). To

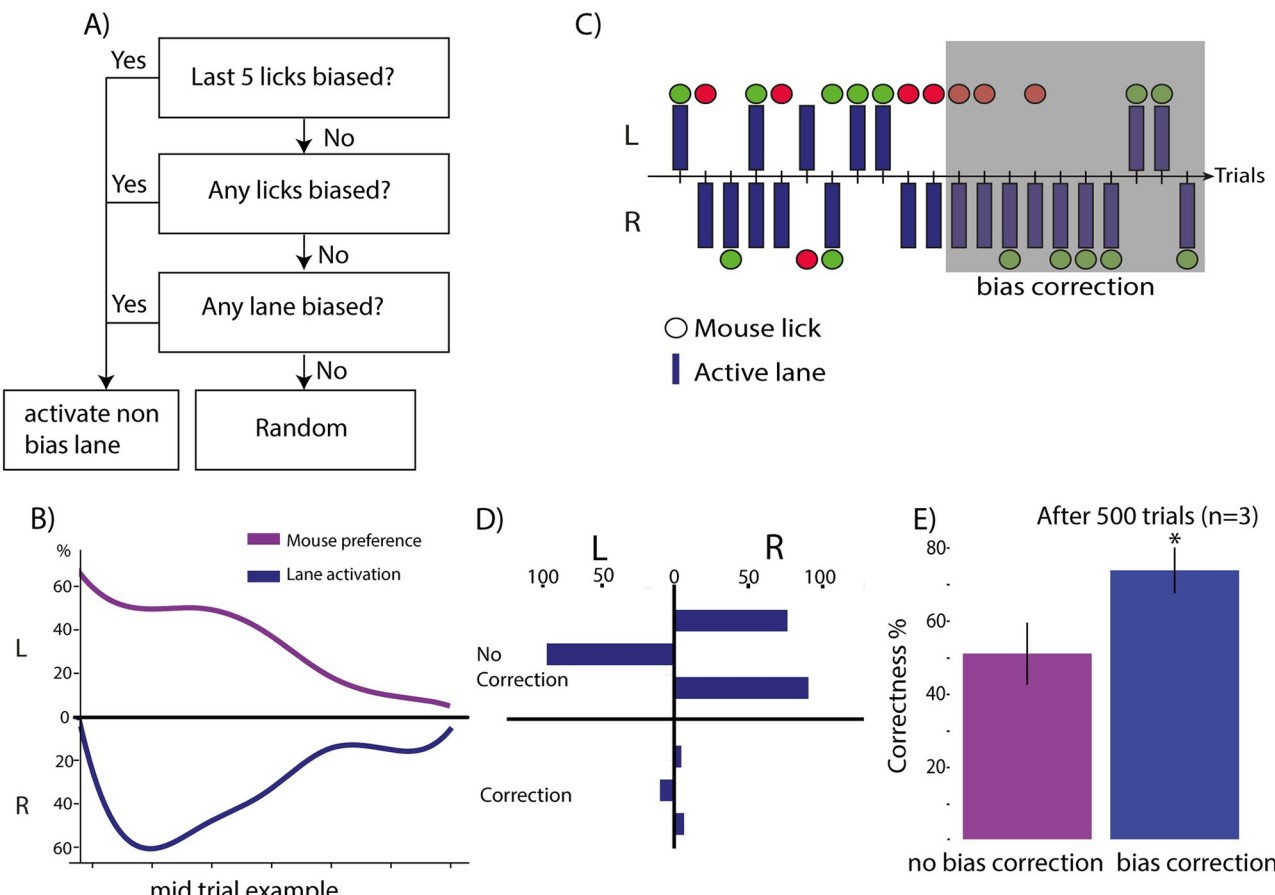

**Fig 3. Bias countering algorithm.** A) Schematic of the decision points in forcing a change in the preference of the animal. If there is no bias, a randomized trial is initiated. B) If the animal is biased, a corrective set of trials is initiated, resulting in the increased activation of the neglected lane. If this is not fruitful, only the neglected lane will be activated to push the mouse into exploring both lanes. The next decision point is when mice lick. If mice licked a port 4 times in a row, a bias correction algorithm was instituted. C) The bias correction had the desired effect, as shown in this example from a single mouse. D) Mice were split into groups trained with bias correction and without bias correction, and the average effects for the two groups are shown here. E) In addition to abolishing the bias, the bias correction forces mice to learn. Their performance improves. P value < 0.05 via a 2 tailed t test.

confirm that the bias correction was effective, we split mice in two groups of 3 mice. One group was trained with the bias correction and the other without bias correction. Bias correction was effective, it improved performance of the mice (Fig 3D and 3E).

## Assessment of training and behavior

To assess whether home-cage training significantly changed the rate of learning, we measured performance (percentage correct) and the average speed of movement. To address whether head fixation fundamentally alters mouse behavior, we used the same mice first in the home-cage system then in the Air-Track. This approach makes it possible to assess the effect of prior training on learning the behavior in the Air-track and provides an opportunity to assess the effect of head fixation on behavior mice in the Y-maze when freely moving, and when head fixed in the Air-track.

## Results

To examine the effect of head fixation on behavior, we compared the behavior of mice performing the same behavior—a 2AFC y-maze task—under freely moving and head fixed conditions and tested whether animals could transfer knowledge learned under freely moving conditions to the head fixed version of the same task. The y-maze floating platform of the Air-Track system was transferred to the home-cage and mice were trained on the same maze in head fixed and freely moving conditions. We compared three different conditions: one cohort of mice was trained exclusively under head fixation, another cohort was trained under freely-moving conditions in their home-cage and a third cohort of mice was pre-trained in their home-cage and transferred to the same task under head fixation. Overall, when we compared the behaviors mice exhibit when freely moving and head fixed in the Air-track, the behaviors were comparable. One constraint imposed by head fixation is that mice have to propel themselves forward and backward without turning their body around. It was possible that in the freely moving condition, mice would always turn around. But even when freely moving in the home-cage, mice quickly learned that it was better to back out of the maze and not turn around. Space constraints in the home-cage Y maze made it difficult for mice to turn around (S1 Fig). Next we quantified movement speed as mice moved through the maze in the home-cage and in the Airtrack (Fig 4A and 4B). Naïve, freely moving mice moved the fastest through the maze and naive head fixed mice moved the slowest. But pre-training in the home-cage dramatically changed movement speed. Pre-trained head fixed mice move at a similar speed as freely moving mice, and move faster than untrained naïve head fixed mice. When mice were head fixed in the Airtrack, the weight of the airtrack was a key variable that affected the speed of their movement: the heavier the maze, the slower the movement of head fixed mice, for both forward or backward movements. The Y maze used for this study was smaller and lighter than the plus maze we used previously ([25]; previous, 30 cm, 160 g; current 15 cm, 75 g).

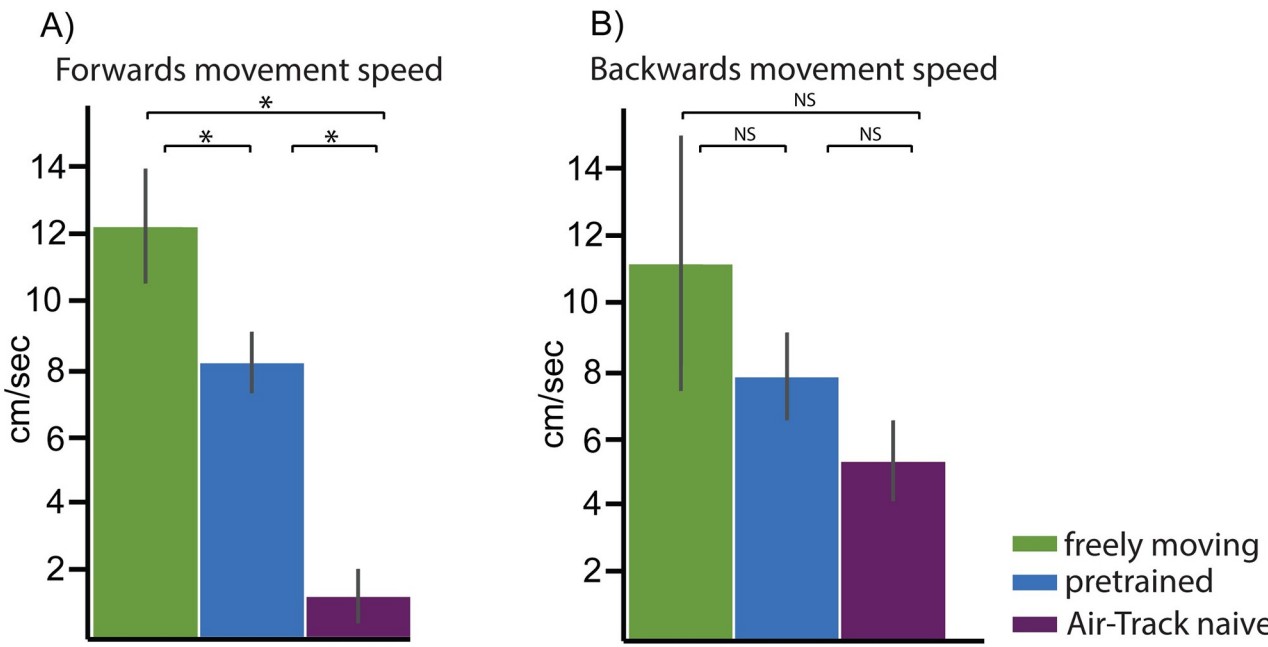

**Fig 4. Impact of Air-Track on behavioral parameters.** Movement speed in forwards (A) and backwards direction (B) for the three conditions. * p<0.05, NS p>0.05 two tailed t test.

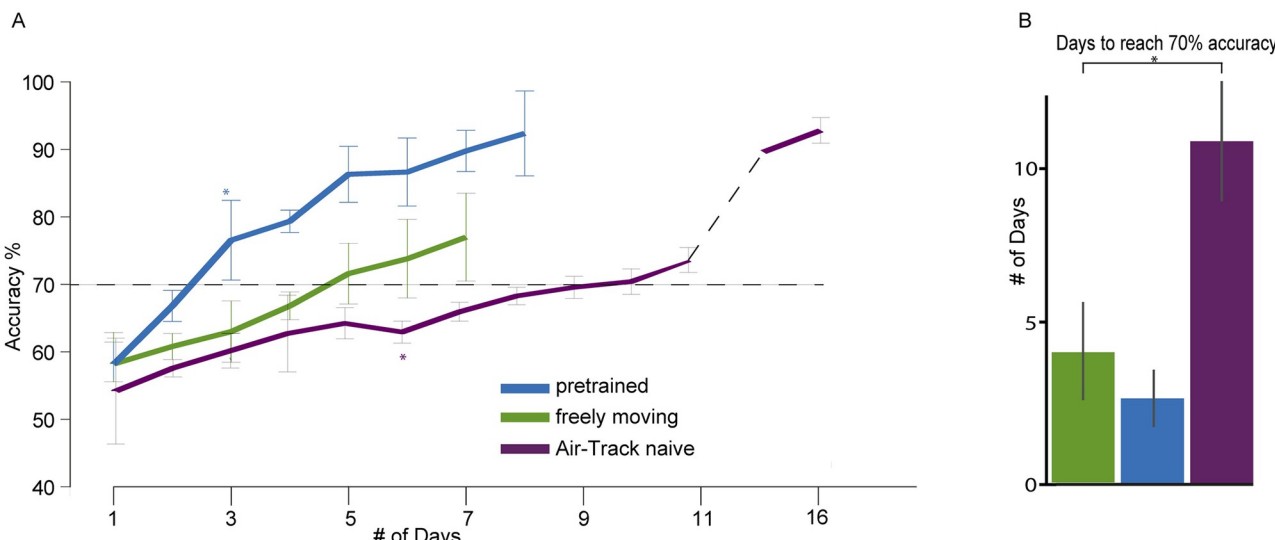

**Fig 5. Head fixed learning improves after pre-training in freely-moving environment.** Comparison of the freely moving, Air-Track naive and Air-Track pre-trained mice in the task. A) Performance of mice (% correct trials) for freely moving, Air-track control (Air-track naive) and Air-track with pre-training (Air-track pre-training). * $p<0.05$, NS $p>0.05$ Mann Whitney U test, $p < 0.05$ B) Time to reach performance criterion (70% success)* $p<0.05$, NS $p>0.05$ two tailed t test).

Finally, we counted the number of lanes mice entered in the course of performing the task and found that there was no significant effect of head fixation on the number of lanes mice entered (S1 Fig).

Next, we examined the effect of pre-training on speed of learning the two alternative forced-choice (2AFC) task. Freely moving mice, trained on the Y-maze task learned much faster than head fixed mice (Fig 5A, green and purple respectively), reaching 70% success criterion on average in 4 +/- 2 days versus 9 +/- 4 days. Mice that were head fixed on the Y-maze after being trained under freely-moving conditions, initially performed at levels equivalent to naive-head fixed mice (Fig 5A, blue). However, they quickly increased their performance, reaching the 70% success criterion in 3 +/- 1 days. The naive-head fixed mice reached criterion performance at the slowest pace, needing 10 days of training. The learning trajectory of pre-trained head fixed mice was the fastest of any category (Fig 5B).

Pre-training freely moving mice (n = 5) on the Air-Track significantly reduced both the total time needed to reach criterion and the number of trials it took to reach criterion. It took mice fewer days to reach 70–85% correct levels if they had been trained in the home-cage first (Mann Whitney U test, $p < 0.05$). These effects of pre-training head fixed mice were plotted as the number of hours it took to train a single mouse to criterion levels, i.e. 70% percent correct (Fig 6A). Pre-training reduced the time needed for mice to learn the task to criterion levels by 70%. Thus, while every mouse required one hour in the Air-Track per training session per day, the pre trained mice required fewer days to "re-learn" the task when they were head fixed. The time saving aspect of the home-cage automation becomes apparent when considering a cohort of 5 or 10 mice (Fig 6B). Assuming 5 mice are required for the experiment, using the home-cage training system reduces the total training time by 31 hours. Note that the total time investment in hours per mouse for a single mouse is not very different in the two cases, but as the cohort of animals being trained increases from 1 to 5 to 10, the difference in training time becomes substantial (Fig 6C).

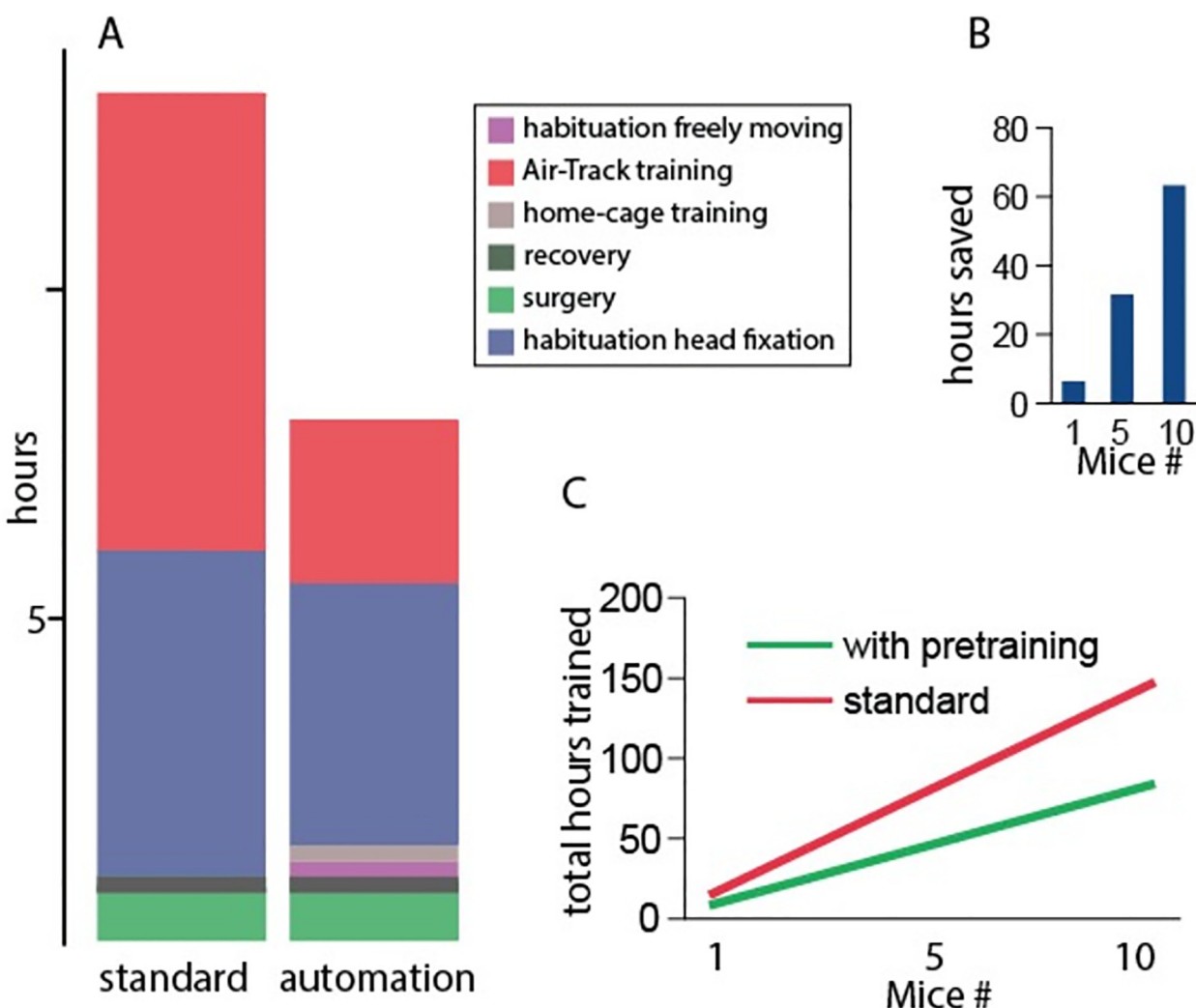

**Fig 6. Active time investment to reach criterion levels.** A) The total time in hours the researcher spends training mice is less due to the home-cage training (left bar). The biggest effect of home-cage training is the reduction in time taken during learning of the task in the Airtrack. Habituation to the freely moving environment and home-cage training are unique to automation. Since those two phases are done in the automated home-cage it requires only a few minutes per day to check on the mice and the rest is done automatically. B) The time saving aspect in relation to home-cage training. C) Comparison of the total time investment in hours. As the number of mice being trained increases, there is a substantial effect of home-cage training on time needed to train mice.

## Discussion

With this work, we show that when trained in a two choice auditory visual spatial discrimination task mice transfer what they learn while freely moving to the head fixed condition. This work also demonstrates that while most aspects of behavior are normal under head fixation, the speed with which mice navigate the floating Air-Track maze can be affected by head fixation.

A fundamental problem faced by all systems neuroscientists is that training mice is a burden and takes a lot of energy and time. As the complexity of tasks we use with mice has increased, it has become imperative that we achieve faster training times and reduce the stress on mice. Here we began this process by developing home-cage training, i.e. training mice in their home-cage to perform a two-choice discrimination task. Our work shows that mice learn in

the home-cage, they get accustomed to the apparatus and can learn a portion of the task while freely moving. Subsequently when they are head fixed, mice reach criterion levels faster if they have been trained in the home-cage. While the reduction in training time is an improvement over standard training of head fixed mice, setting up home-cage training requires a lot of effort. It requires setting up a task that is compatible for the home-cage. Ultimately, it is necessary to develop methods for monitoring the trial by trial behavior in the home-cage, and it requires tracking the location and performance of the animal or animals while they are freely moving. Each behavioral task requires its own unique home-cage design and setup.

## Comparison to other home-cage training and task designs

Training mice in their home-cage has been a central goal of earlier work of the lab of Murphy and colleagues (2018; [30]). While their work did not address whether mice learned faster when trained while freely moving, their work showed that mice could be trained to learn all aspects of behavior in their home-cages. The design of their home cage included several additional features that we did not deploy, including self-head fixation and imaging neural activity in the home-cage after mice self-head fix. One additional difference between their task design and ours, was that in our design, mice could perform the task and have access to water rewards for only 1 hour each day. In the earlier work from Murphy, they used two groups, in the first group mice could perform24/7 hrs and over 3 weeks of training mice achieved 55% success rates and the second smaller group had access for 7hrs per day with 35% success rate in a cue-based licking task ([30]). In another study, where mice could perform the task at any time, mice reached criterion levels within 31–37 days (T [31]). In our version of the home cage training, mice had access to water for an hour each day, and in this hour they had to perform the task. Limiting the task to an hour, and limiting the availability of water reward to a single hour, focused mice on learning and performing the task. In our initial experiments (not shown) where mice had 24/7 access to the training area for 3 days and could perform the task at any time, mice showed no evidence that they were learning the task. When we changed the design, mice learned the task and performance reached criterion within 7 days of training.

## Effect of head fixation on behavior

Our main concern was to examine whether mice expressed the same behavioral strategies when they were freely moving or head fixed. One of the unique features of the plus and y-mazes used here were that mice had to move the maze forward and backwards ([25]). While the Air-Track enables the simulation of the animal's natural environment with behavior, it was not clear whether freely moving mice would walk backwards in lanes, or would try to turn around and walk forward to exit a lane. Here we show that when the space in a lane was constrained, i.e. the top was closed, even freely moving mice naturally tend to walk backwards to exit the lane. From these observations, we can conclude that the behaviors observed in the Air-Track are qualitatively similar to those seen in freely moving mice. Another concern with head fixation is that it affects the normal pattern of coordinated behavior. For example, walking is normally associated with head movement, and therefore under head fixation mice need to learn to walk without head movement. Head fixation therefore might alter the animal's natural gait. Indeed, our work suggests that head fixed mice move more slowly through the maze. Part of the reduction in speed arises from the platform itself, since moving is initiated by the mouse translating the position of the platform moving by translating the position of the platform requires a different kind of force than normal walking, and this adds to the time mice need to move through the maze. Head fixation obviously prevents head movement, and can therefore increase stress in animals ([34]). In rodents, head movements are correlated with whisker

movements ([35–44]). In many species including rodents, primates and in humans, head movements are also linked to eye movement ([45–50]). Thus, in addition to preventing the expression of some motor sequences and potentially increasing stress levels, head fixation can change some behaviors. Additionally, when rodents are head fixed they only make large conjugate eye movements but when they are freely moving they make small disconjugate eye movements and large conjugate eye movements ([49, 50]). Nevertheless, the advantages of head fixation even in these conditions are obvious: Monitoring eye movement in freely moving mice requires head mounted cameras. It is much easier to monitor eye movement with cameras in the head fixed mice than in freely moving animal. Additionally, head fixation has been shown to improve performance in some behaviors. When mice are trained to reach for a water droplet, head fixed mice are more focused, reach criterion performance faster, succeed at more trials, even though head fixation reduces the range of movement ([29]). A growing number of labs are recognizing the need to develop home-cage training because it can speed up training in complex tasks. The ability to monitor mouse behavior in the home-cage, and to track the performance of individual mice in the home-cage is a first step. Our study shows that mice can transfer what they learn from one context to the head fixed condition. In the future we plan to have mice self head fix which will trigger the Air-Track to float.

## Supporting information

**S1 Fig. Comparing the different methods to analyze the behavior of mice during their training.** A) Comparing the movement speed of freely moving mice with naive head fixed and pre-trained head fixed mice.* $p < 0.05$, NS $p > 0.05$ via a two tailed T test B) Comparing the lane entries for head fixed and home-cage mice, showed no significance $p > 0.05$ C) Freely moving mice in the home-cage are able to rotate in the lane, but prefer to move backwards out of the lane. With no turning around at day 3.
(TIF)

**S2 Fig. Comparing each mouse in the freely moving and head fixed environment.** Each dot plot represents a single mouse which was trained in the home-cage as well as the Air-Track, as well as a line plot showing the time point each mouse reaches the 70% correctness.
(TIF)

**S3 Fig. Time it takes to train a mouse to reach criterion levels in days.** Representation of the average time in days it takes to train a single mouse, including surgery, recovery, and training in the traditional standard way of training mice for the Air-Track and the automated home-cage training.
(TIF)

## Acknowledgments

We thank the Charité Workshop for technical assistance, especially Alexander Schill, Jan-Erik Ode and Daniel Deblitz. We thank Drs Tim Zolnik and Malinda Tantirigama for their useful suggestions and comments on earlier versions of this manuscript.

## Author Contributions

**Conceptualization:** Anna Nasr, Sina E. Dominiak, Mostafa A. Nashaat, Robert N. S. Sachdev, Matthew E. Larkum.

**Formal analysis:** Keisuke Sehara.

**Funding acquisition:** Anna Nasr, Matthew E. Larkum.

**Investigation:** Anna Nasr, Sina E. Dominiak.

**Methodology:** Keisuke Sehara.

**Project administration:** Anna Nasr.

**Software:** Anna Nasr.

**Supervision:** Robert N. S. Sachdev, Matthew E. Larkum.

**Visualization:** Keisuke Sehara.

**Writing – original draft:** Anna Nasr, Robert N. S. Sachdev, Matthew E. Larkum.

**Writing – review & editing:** Anna Nasr, Keisuke Sehara, Robert N. S. Sachdev, Matthew E. Larkum.

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
