## [Decision Letter · Decision Letter 0]

7 Jun 2022

PONE-D-22-05001Efficient training approaches for optimizing behavioral performance and reducing head fixation timePLOS ONE

Dear Dr. Nasr,

Thank you for submitting your manuscript to PLOS ONE. After careful consideration, we feel that it has merit but does not fully meet PLOS ONE’s publication criteria as it currently stands. Therefore, we invite you to submit a revised version of the manuscript that addresses the points raised during the review process.

 Both reviewers agree on the merits of the manuscript. However, reviewer 1 raises a significant number of questions, concerns and suggestions which do not require new experiments to be addressed, but will expand the analysis. In addition clarifications are asked, especially by reviewer 2 regarding references and a number of suggestions are made by both reviewers aiming at improving clarity. Please thoroughly address all of these concerns and suggestions.

We look forward to receiving your revised manuscript.

Kind regards,

Efthimios M. C. Skoulakis, PhD

Academic Editor

PLOS ONE

Journal Requirements:

The following funding sources have supported this project: (1) Deutsche Forschungsgemeinschaft (DFG), Grant Nos. 246731133, 250048060 and 267823436 to ML; (2) DFG Project number 327654276 – SFB 1315 to ML; (3) European Commission Horizon 2020 Research And Innovation Program and Euratom Research and Training Program 2014–2018 (under grant agreement No. 670118 to ML); (4) Human Brain Project, EU Commission Grant 720270 (SGA1), 785907 (SGA2) and 945539 (SGA3) to ML; (5) Einstein Foundation Berlin EVF-2017-363 to ML.

6. Please amend your list of authors on the manuscript to ensure that each author is linked to an affiliation. Authors’ affiliations should reflect the institution where the work was done (if authors moved subsequently, you can also list the new affiliation stating “current affiliation:….” as necessary).’

Reviewers' comments:

Reviewer's Responses to Questions

**Comments to the Author**

1. Is the manuscript technically sound, and do the data support the conclusions?

Reviewer #1: Yes

Reviewer #2: Yes

2. Has the statistical analysis been performed appropriately and rigorously? 

Reviewer #1: Yes

Reviewer #2: Yes

3. Have the authors made all data underlying the findings in their manuscript fully available?

Reviewer #1: Yes

Reviewer #2: No

4. Is the manuscript presented in an intelligible fashion and written in standard English?

Reviewer #1: Yes

Reviewer #2: Yes

5. Review Comments to the Author

Reviewer #1: The authors in this paper investigate how pretraining with freely moving animals can be used to improve the learning rates in head-immobilized tasks. Specifically, the authors utilize a unique two-choice discrimination task with a Y maze that can be used with both freely moving and head-immobilized animals. They use this apparatus to compare the learning rates of three different groups of mice: a) Ones in the freely moving task b) Naïve ones while head-fixed and c) head-fixed animals that have been pretrained in the freely moving task. The authors find a significant decrease in training required for the last group of animals.

The question addressed in this paper is of great importance to the field. Given the requirement of many advanced recording techniques for head-immobilized animals together with complex behavior, their results provide support for using pre-training with complex behavioral tasks that the animals can generalize while head-posted. Moreover, their Y-maze design is easy to generalize over and can also be useful for many behavioral questions in the field.

Even though the authors try to address a relevant question with a good behavioral task that can be applied in both freely moving and head-immobilized animals, I feel like it needs a bit more work to become much more useful to the field.

Major:

1. Quantification and comparison of the behavioral patterns between the freely moving and head-posted version of the task is lacking a bit. Given the very short length of the results section the authors could augment them by looking into different parts of the behavior for already collected data. For example:

a. As the authors have used the same animals between the two conditions, it would be interesting to know how the animal variability looks in such a task. For example, the authors could investigate the difference between the performance criterion for the two tasks across animals (each axis represents the performance criterion for the two tasks and for each animal in a scatter plot).

b. Animals can perform complex movements in both tasks but only the movement speed is compared. It would be easy to compare other aspects of the movement such as distance traveled, rotations or even # of lane entries.

c. It would be important to know the maximum performance that these animals can reach in each version of the task. Do mice that are trained on the air platform without pretraining reach similar levels of performance to pretrained animals?

2. Why is the performance of the freely moving task lower than the head-posted task? Is it the extra training the animals received or is it because of the “animals being more focused” as they mention in the discussion? In the discussion the authors need to clearly mention this. Is it also possible that animals need more days under water restriction to reach stable motivation levels as has been shown in rats (Vasilev et al. 2021)?

3. In Figure 6A I probably miss something. For the home cage training to reach criterion the animals required about 7 days of training which is >10hours per animal. In the bar plot of Fig. 6A the home-cage training takes <1hour. Why is that?

4. Is the backwards movement defined as a reference to the platform or the animals? From the discussion you mention freely moving animals moving backwards under certain circumstances, but I am surprised that animals can move 10cm/sec backwards. If is relative to the platform (so animals rotate while head-posted and then move backwards) would make sense.

5. I don’t think that the authors provide evidence that “most-natural behaviors are unaffected”, as I find this is too general of a statement. Maybe the performance in this task is unaffected, but I would suggest the authors tone down this statement. A nice experiment that would offer support for such a claim would be to train animals on the air-platform and then see if it improves the performance in a freely moving task. I don’t know if the camera used had enough resolution, but it would be also interesting to check for snout /whisker movements as they have been shown to correlate with sampling of peripersonal space (Kurnikova et al. 2017). Even though that might be beyond the scope of this work, it would offer support for this head-immobilized task being a “natural behavior”. Do you expect to have such behaviors or to have them disappear like the disconjugate eye movements?

Minor:

1. A schematic representation of the various timing of training events would be very helpful, since this information is not in one place in the text and not very easy to find.

2. Why is there a different criterion applied in figures 5 and 6 (70% vs 75%)?

3. Text edit: “The mice were acclimated to being handled, and held by the head post during which they obtained a reward when the head post was manually held in place by the experimenter” would be better as: “The mice were acclimated to being handled, during which the animals obtained a reward while the head post was manually held in place by the experimenter”

4. I would replace the “improve the learning curve” with “improve the learning rate”.

5. “To minimize the time between home-cage training and Air-Track training as well as outside factors did all the mice undergo the surgery at the beginning of the experiment. “ -> “… outside factors, all mice underwent surgery …”

6. Two paragraphs after “Daily training routine: Common to both home-cage and head fixed training” are repeated.

7. Figure 3 should be used in the bias describing paragraph not 4.

8. “To answer the question regarding the naturality of the head fixed behavior, were the same mice, first used in the home-cage system and then transferred to the Air-Track” Sentence not clear.

References:

Kurnikova et al. Coordination of Orofacial Motor Actions into Exploratory Behavior by Rat. Current Biology 2017, 27 (5), 688–696.

Vasilev, D. et al. Three Water Restriction Schedules Used in Rodent Behavioral Tasks Transiently Impair Growth and Differentially Evoke a Stress Hormone Response without Causing Dehydration. eNeuro 8, (2021).

Reviewer #2: The author's compare training of mice in either head fixed or freely moving task and show that pre-training for headfixed tasks has some benefit, furthermore performance is not greatly compromised by headfixation. This work is important for any labs developing high-throughput means of assessment.

"This is in contrast to the earlier work where mice had access to water for 7 hrs each day, and

over 3 weeks of training mice only achieved 55% success rates in a cue-based licking task

(Murphy 2020)." Perhaps it was not clear but in Murphy et al. 2020 mice were trained both in a 7 h a day mode and 24/7 (44 mice 24/7) and 8 females 7 h a day with similar performance.

need to state that pixycam only works with visible light (colored).

Gallinares et al should be Galiñanes et al. and is NOT in the reference list

other references are out of order or missing Murphy et al. 2017

please carefully check the manscript and references for accuracy

https://gin.g-node.org/nasra/Prior-experience-accelerates-training-of-head-fixed-miceon-

a-floating-platform the data link did not work please fix

6. PLOS authors have the option to publish the peer review history of their article (what does this mean?). If published, this will include your full peer review and any attached files.

Reviewer #1: No

Reviewer #2: No

---

## [Author Response · Author response to Decision Letter 0]

27 Jul 2022

The document containing all this commets as well as my responses are in the document Response to Reviewers:

1. Please ensure that your manuscript meets PLOS ONE's style requirements, including those for file naming. The PLOS ONE style templates can be found at https://journals.plos.org/plosone/s/file?id=wjVg/PLOSOne_formatting_sample_main_body.pdf andhttps://journals.plos.org/plosone/s/file?id=ba62/PLOSOne_formatting_sample_title_authors_affiliations.pdf

We followed the guidelines

We used the template

We will correct the mistake.

The following funding sources have supported this project: (1) Deutsche Forschungsgemeinschaft (DFG), Grant Nos. 246731133, 250048060 and 267823436 to ML; (2) DFG Project number 327654276 – SFB 1315 to ML; (3) European Commission Horizon 2020 Research And Innovation Program and Euratom Research and Training Program 2014–2018 (under grant agreement No. 670118 to ML); (4) Human Brain Project, EU Commission Grant 720270 (SGA1), 785907 (SGA2) and 945539 (SGA3) to ML; (5) Einstein Foundation Berlin EVF-2017-363 to ML.

The funders had no role in study design, data collection and analysis, decision to publish, or preparation of the manuscript

The repository we have provided is the following:

https://gin.g-node.org/nasra/Prior-experience-accelerates-training-of-head-fixed-mice-on-a-floating-platform.git

We will provide the relevant data in this repository once the manuscript has been accepted for publication.

6. Please amend your list of authors on the manuscript to ensure that each author is linked to an affiliation. Authors’ affiliations should reflect the institution where the work was done (if authors moved subsequently, you can also list the new affiliation stating “current affiliation:….” as necessary).’

We have updated the list of authors.

Reviewers' comments:

Reviewer's Responses to Questions

Comments to the Author

1. Is the manuscript technically sound, and do the data support the conclusions?

Reviewer #1: Yes

Reviewer #2: Yes

2. Has the statistical analysis been performed appropriately and rigorously?

Reviewer #1: Yes

Reviewer #2: Yes

3. Have the authors made all data underlying the findings in their manuscript fully available?

Reviewer #1: Yes

Reviewer #2: No

https://gin.g-node.org/nasra/Prior-experience-accelerates-training-of-head-fixed-mice-on-a-floating-platform.git

4. Is the manuscript presented in an intelligible fashion and written in standard English?

PLOS ONE does not copy edit accepted manuscripts, so the language in submitted articles must be clear, correct, and unambiguous. Any typographical or grammatical errors should be corrected at revision, so please note any specific errors here.

Reviewer #1: Yes

Reviewer #2: Yes

5. Review Comments to the Author

Reviewer #1: The authors in this paper investigate how pretraining with freely moving animals can be used to improve the learning rates in head-immobilized tasks. Specifically, the authors utilize a unique two-choice discrimination task with a Y maze that can be used with both freely moving and head-immobilized animals. They use this apparatus to compare the learning rates of three different groups of mice: a) Ones in the freely moving task b) Naïve ones while head-fixed and c) head-fixed animals that have been pre-trained in the freely moving task. The authors find a significant decrease in training required for the last group of animals.

The question addressed in this paper is of great importance to the field. Given the requirement of many advanced recording techniques for head-immobilized animals together with complex behavior, their results provide support for using pre-training with complex behavioral tasks that the animals can generalize while head-posted. Moreover, their Y-maze design is easy to generalize over and can also be useful for many behavioral questions in the field.

Even though the authors try to address a relevant question with a good behavioral task that can be applied in both freely moving and head-immobilized animals, I feel like it needs a bit more work to become much more useful to the field.

Major:

1. Quantification and comparison of the behavioral patterns between the freely moving and head-posted version of the task is lacking a bit. Given the very short length of the results section the authors could augment them by looking into different parts of the behavior for already collected data. For example:

a. As the authors have used the same animals between the two conditions, it would be interesting to know how the animal variability looks in such a task. For example, the authors could investigate the difference between the performance criterion for the two tasks across animals (each axis represents the performance criterion for the two tasks and for each animal in a scatter plot).

We thank the reviewer for this suggestion and have prepared plots for each mouse, in Supplementary Figure 2. 

b. Animals can perform complex movements in both tasks, but only the movement speed is compared. It would be easy to compare other aspects of the movement such as distance traveled, rotations or even # of lane entries.

We agree and have generated some comparative figures (Supplementary Figure 2). One comparison that we have not made is of distance traveled. Mice in the home-cage can leave the training area and move around freely, so the travel distance is not comparable to the limited environment of the Air-Track.

c. It would be important to know the maximum performance that these animals can reach in each version of the task. Do mice that are trained on the air platform without pretraining reach similar levels of performance to pretrained animals?

Thank you for the suggestion, we added more details to Figure 5 showing the final performance of the naïve Air-Track mice. This figure shows the total time it takes for mice to reach the same level as the pre-trained mice.

2. Why is the performance of the freely moving task lower than the head-posted task? Is it the extra training the animals received, or is it because of the “animals being more focused” as they mention in the discussion? In the discussion, the authors need to clearly mention this. Is it also possible that animals need more days under water restriction to reach stable motivation levels, as has been shown in rats (Vasilev et al. 2021)?

The pre-trained mice transfer their knowledge of the freely moving task to the Air-Track, which results in a continuation of the learning curve. The drop in performance in the first 3 days is due to the fact that even after being habituated to head fixation, and mice still need to learn to move the Air-Track and need to be motivated to perform the task.

3. In Figure 6A I probably miss something. For the home cage training to reach criterion the animals required about 7 days of training which is >10hours per animal. In the bar plot of Fig. 6A the home-cage training takes <1hour. Why is that?

We have rewritten and clarified Figure 6 legend. Figure 6. Active time investment to reach criterion levels. A) The total time in hours the researcher spends training mice is less due to the home cage training (left bar). The biggest effect of home-cage training is the reduction in time taken during learning of the task in the Airtrack. Habituation to the freely moving environment and home-cage training are unique to automation. Since those two phases are done in the automated home-cage it requires only a few minutes per day to check on the mice and the rest is done automatically. B) The time saving aspect in relation to home cage training. C) Comparison of the total time investment in hours. As the number of mice being trained increases, there is a substantial effect of home-cage training on time needed to train mice.

4. Is the backwards movement defined as a reference to the platform or the animals? From the discussion, you mention freely moving animals moving backwards under certain circumstances, but I am surprised that animals can move 10cm/sec backwards. If is relative to the platform (so animals rotate while head-posted and then move backwards) would make sense.

The movement speed was calculated by using a basler high speed camera and the software ZRView. This allowed us to count the frames between the end of the lane and the center of the Y maze. Using this method, we obtained the number of frames it takes to move from the end of the lane back to the center. The number of frames divided by the frame rate resulted in a cm/sec value. 

5. I don’t think that the authors provide evidence that “most-natural behaviors are unaffected”, as I find this is too general of a statement. Maybe the performance in this task is unaffected, but I would suggest the authors tone down this statement. A nice experiment that would offer support for such a claim would be to train animals on the air-platform and then see if it improves the performance in a freely moving task. I don’t know if the camera used had enough resolution, but it would be also interesting to check for snout /whisker movements as they have been shown to correlate with sampling of peripersonal space (Kurnikova et al. 2017). Even though that might be beyond the scope of this work, it would offer support for this head-immobilized task being a “natural behavior”. Do you expect to have such behaviors or to have them disappear, like the disconjugate eye movements?

We agree with the reviewer, and have rewritten the paragraph. “We conclude that home-cage pre-training improves learning performance of head fixed mice and that while head fixation obviously limits some aspects of movement, the patterns of behavior observed in head fixed and freely moving mice are similar.” 

Our cameras did not have the resolution to track whiskers, and were not in a position to track eye movement. We do expect that in the course of behavior, in freely moving animals, eye movement would be disconjugate, and both whisking and eye movement would be linked to head movement. We have tracked both eyes and whiskers in mice head fixed in a plus maze and show that eye movement is conjugate and related to whisker movement(Bergmann et., al 2021). They concluded that not only do head-fixed mice move their eyes, but saccadic eye movement is coordinated with the asymmetric positioning of the whiskers.

Minor:

1. A schematic representation of the various timing of training events would be very helpful, since this information is not in one place in the text and not very easy to find.

Thank you for pointing this out. We have fixed this and added a supplementary figure (Supplementary Figure 3). 

2. Why is there a different criterion applied in figures 5 and 6 (70% vs 75%)?

Thank you for noticing this was a typo, and the criterion is now uniformly 70%.

3. Text edit: “The mice were acclimated to being handled, and held by the head post during which they obtained a reward when the head post was manually held in place by the experimenter” would be better as: “The mice were acclimated to being handled, during which the animals obtained a reward while the head post was manually held in place by the experimenter”

We have rewritten this sentence: “The mice were acclimated to being handled, during which the animals obtained a reward while the head post was manually held in place by the experimenter.”

4. I would replace the “improve the learning curve” with “improve the learning rate”.

We implemented this change: “The fundamental issue which we tried to address here was to improve the learning rate of mice and in the course of doing so we also examined how and whether head fixation changed behavior. Our work suggests that pre-training animals under freely-moving conditions reduced the total time the animals spent learning while head fixed and that head fixation does not trigger aberrant behavior. ”

5. “To minimize the time between home-cage training and Air-Track training as well as outside factors did all the mice undergo the surgery at the beginning of the experiment. “ -> “… outside factors, all mice underwent surgery …”

We have rewritten this sentence.:” To minimize the time between home-cage training and Air-Track training as well as outside factors, all mice underwent surgery at the beginning of the experiment.” 

6. Two paragraphs after “Daily training routine: Common to both home-cage and head fixed training” are repeated.

Thank you for pointing this out, we removed the duplicate paragraph.

7. Figure 3 should be used in the bias describing paragraph, not 4.

We have corrected this error. 

8. “To answer the question regarding the naturality of the head fixed behavior, were the same mice, first used in the home-cage system and then transferred to the Air-Track” Sentence not clear.

We have rewritten this paragraph: To assess whether home-cage training significantly changed the rate of learning, we measured performance (percentage correct) and the average speed of movement. To address whether head fixation fundamentally alters mouse behavior, we used the same mice first in the home-cage system then in the Air-Track. This approach makes it possible to assess the effect of prior training on learning the behavior in the Air-track and provides an opportunity to assess the effect of head fixation on behavior mice in the Y-maze when freely moving, and when head fixed in the Air-track. 

References:

Kurnikova et al. Coordination of Orofacial Motor Actions into Exploratory Behavior by Rat. Current Biology 2017, 27 (5), 688–696.

Vasilev, D. et al. Three Water Restriction Schedules Used in Rodent Behavioral Tasks Transiently Impair Growth and Differentially Evoke a Stress Hormone Response without Causing Dehydration. eNeuro 8, (2021).

Reviewer #2: The author's compare training of mice in either head fixed or freely moving task and show that pre-training for headfixed tasks has some benefit, furthermore performance is not greatly compromised by headfixation. This work is important for any labs developing high-throughput means of assessment.

"This is in contrast to the earlier work where mice had access to water for 7 hrs each day, and over 3 weeks of training mice only achieved 55% success rates in a cue-based licking task (Murphy 2020)." Perhaps it was not clear but in Murphy et al. 2020 mice were trained both in a 7 h a day mode and 24/7 (44 mice 24/7) and 8 females 7 h a day with similar performance.

We have rewritten the entire paragraph:Training mice in their home-cage has been a central goal of earlier work of the lab of Murphy and colleagues (2018; 2020). While their work did not address whether mice learned faster when trained while freely moving, their work showed that mice could be trained to learn all aspects of behavior in their home-cages. The design of their home cage included several additional features that we did not deploy, including self-head fixation and imaging neural activity in the home-cage after mice self-head fix. One additional difference between their task design and ours, was that in our design, mice could perform the task and have access to water rewards for only 1 hour each day. In the earlier work from Murphy, they used two groups, in the first group mice could perform24/7 hrs and over 3 weeks of training mice achieved 55% success rates and the second smaller group had access for 7hrs per day with 35% success rate in a cue-based licking task (Murphy 2020). In another study, where mice could perform the task at any time, mice reached criterion levels within 31-37 days (Tejapratap BolluGoldberg et al., 2018). In our version of the home cage training, mice had access to water for an hour each day, and in this hour they had to perform the task. Limiting the task to an hour, and limiting the availability of water reward to a single hour, focused mice on learning and performing the task. In our initial experiments (not shown) where mice had 24/7 access to the training area for 3 days and could perform the task at any time, mice showed no evidence that they were learning the task. When we changed the design, mice learned the task and performance reached criterion within 7 days of training. 

Need to state that pixycam only works with visible light (colored).

We have rewritten this sentence to: When a mouse entered the maze, the identity of the mouse was checked via the color tracking algorithm and Pixy camera (see below) deployed to track colors (under visible light) painted on the head post of each animal.

Gallinares et al should be Galiñanes et al. and is NOT in the reference list

We have corrected this error.

Other references are out of order or missing Murphy et al. 2017

The reference order has been fixed, and the missing references have been added.

please carefully check the manscript and references for accuracy

Thank you very much for pointing it out. We went over all references and corrected the inaccuracies as well as added more references.

https://gin.g-node.org/nasra/Prior-experience-accelerates-training-of-head-fixed-miceon-

a-floating-platform

Please excuse the inconvenience. The link should work now. 

https://gin.g-node.org/nasra/Prior-experience-accelerates-training-of-head-fixed-mice-on-a-floating-platform.git

6. PLOS authors have the option to publish the peer review history of their article (what does this mean?). If published, this will include your full peer review and any attached files.

Do you want your identity to be public for this peer review? For information about this choice, including consent withdrawal, please see our Privacy Policy.

Reviewer #1: No

Reviewer #2: No

---

## [Decision Letter · Decision Letter 1]

16 Aug 2022

PONE-D-22-05001R1Efficient training approaches for optimizing behavioral performance and reducing head fixation timePLOS ONE

Dear Dr. Nasr,

Thank you for submitting your manuscript to PLOS ONE. After careful consideration, we feel that it has merit but does not fully meet PLOS ONE’s publication criteria as it currently stands. Therefore, we invite you to submit a revised version of the manuscript that addresses the points raised during the review process.

Though most reviewer's comments were addressed satisfactorily, there are a few more technical issues raised by reviewer 1  and detailed below that need to be addressed before teh manuscript is ready for publication  ==============================

We look forward to receiving your revised manuscript.

Kind regards,

Efthimios M. C. Skoulakis, PhD

Academic Editor

PLOS ONE

Journal Requirements:

Reviewers' comments:

Reviewer's Responses to Questions

**Comments to the Author**

1. If the authors have adequately addressed your comments raised in a previous round of review and you feel that this manuscript is now acceptable for publication, you may indicate that here to bypass the “Comments to the Author” section, enter your conflict of interest statement in the “Confidential to Editor” section, and submit your "Accept" recommendation.

Reviewer #1: All comments have been addressed

Reviewer #2: All comments have been addressed

2. Is the manuscript technically sound, and do the data support the conclusions?

Reviewer #1: Yes

Reviewer #2: Yes

3. Has the statistical analysis been performed appropriately and rigorously? 

Reviewer #1: No

Reviewer #2: Yes

4. Have the authors made all data underlying the findings in their manuscript fully available?

Reviewer #1: No

Reviewer #2: Yes

5. Is the manuscript presented in an intelligible fashion and written in standard English?

Reviewer #1: Yes

Reviewer #2: Yes

6. Review Comments to the Author

Reviewer #1: I would like to thank the authors for addressing all of my concerns.

I only have few minor points:

Fig.3e, Fig4, Fig5, Supp. Fig.1a, would benefit if the statistical tests and the p values are mentioned in the figure legends.

Supp Fig1c : Is the "rotating" bar from Day3 missing? Also are some of the error bars missing?

The response to my point 1a is not exactly what I had in mind (days to criterion for each task in the two axis) but the data in Supp. Fig. 2 do provide the relevant information. For visibility purposes the authors can make that a line plot.

Reviewer #2: no further issues important, nice to have the new citations about training this seems like a good paper they might cite this

Vasilev, D. et al. Three Water Restriction Schedules Used in Rodent Behavioral Tasks Transiently Impair Growth and Differentially Evoke a Stress Hormone Response

without Causing Dehydration. eNeuro 8, (2021)

7. PLOS authors have the option to publish the peer review history of their article (what does this mean?). If published, this will include your full peer review and any attached files.

Reviewer #1: No

Reviewer #2: No

---

## [Author Response · Author response to Decision Letter 1]

26 Sep 2022

To the editor and reviewers. 

The manuscript marked-up copy was written and updated using Google Docs and then saved as docx format. The finished manuscript is written in LaTeX, following the style and formatting guidelines of PLOS ONE. In addition to the changes requested by the reviewers, we made some additional minor changes to improve readability. 

Reviewer #1: I would like to thank the authors for addressing all of my concerns.

I only have few minor points:

Fig.3e, Fig4, Fig5, Supp. Fig.1a, would benefit if the statistical tests and the p values are mentioned in the figure legends.

We added the p values in the description for Fig.3e, Fig4, Fig5, Supp. Fig.1a

Supp Fig1c : Is the "rotating" bar from Day3 missing? Also are some of the error bars missing?

Thank you for your comment. Supp Fig 1 C was in the original draft for the publication, but we removed it, since we did not see a good way to present the data. For day 2 there was no visible error bar because the standard deviation is less than 1%, but we added a minimal error bar for visual representation and for Day 3 there is no bar due to the fact that not a single mouse rotated in the maze after the second day. But we added a highlight at 0 for visualization.

The response to my point 1a is not exactly what I had in mind (days to criterion for each task in the two axis) but the data in Supp. Fig. 2 do provide the relevant information. For visibility purposes, the authors can make that a line plot.

We have added the line plot.

Reviewer #2: no further issues important, nice to have the new citations about training this seems like a good paper they might cite this

 We added the citation:

Vasilev, D. et al. Three Water Restriction Schedules Used in Rodent Behavioral Tasks Transiently Impair Growth and Differentially Evoke a Stress Hormone Response without Causing Dehydration. eNeuro 8, (2021)

---

## [Decision Letter · Decision Letter 2]

10 Oct 2022

Efficient training approaches for optimizing behavioral performance and reducing head fixation time

PONE-D-22-05001R2

Dear Dr. Nasr,

We’re pleased to inform you that your manuscript has been judged scientifically suitable for publication and will be formally accepted for publication once it meets all outstanding technical requirements.

Kind regards,

Efthimios M. C. Skoulakis, PhD

Academic Editor

PLOS ONE

Additional Editor Comments (optional):

Reviewers' comments:

Reviewer's Responses to Questions

**Comments to the Author**

1. If the authors have adequately addressed your comments raised in a previous round of review and you feel that this manuscript is now acceptable for publication, you may indicate that here to bypass the “Comments to the Author” section, enter your conflict of interest statement in the “Confidential to Editor” section, and submit your "Accept" recommendation.

Reviewer #1: All comments have been addressed

2. Is the manuscript technically sound, and do the data support the conclusions?

Reviewer #1: Yes

3. Has the statistical analysis been performed appropriately and rigorously? 

Reviewer #1: Yes

4. Have the authors made all data underlying the findings in their manuscript fully available?

Reviewer #1: Yes

5. Is the manuscript presented in an intelligible fashion and written in standard English?

Reviewer #1: Yes

6. Review Comments to the Author

Reviewer #1: They authors did a good job improving the manuscript. I find that all my comments have been addressed.

7. PLOS authors have the option to publish the peer review history of their article (what does this mean?). If published, this will include your full peer review and any attached files.

Reviewer #1: No
